# Preparation and Evaluation of a Dosage Form for Individualized Administration of Lyophilized Probiotics

**DOI:** 10.3390/pharmaceutics15030910

**Published:** 2023-03-10

**Authors:** Nicole Fülöpová, Natália Chomová, Jan Elbl, Dagmar Mudroňová, Patrik Sivulič, Sylvie Pavloková, Aleš Franc

**Affiliations:** 1Department of Pharmaceutical Technology, Faculty of Pharmacy, Masaryk University, 612 42 Brno, Czech Republic; 2Department of Microbiology and Immunology, University of Veterinary Medicine and Pharmacy, Komenského 73, 041 81 Košice, Slovakia; 3Department of Natural Drugs, Faculty of Pharmacy, Masaryk University, 612 42 Brno, Czech Republic

**Keywords:** lyophilization, antropozoonoses, probiotic bacteria, individual treatment, principal component analysis, viability of bacteria

## Abstract

Probiotics have been used in human and veterinary medicine to increase resistance to pathogens and provide protection against external impacts for many years. Pathogens are often transmitted to humans through animal product consumption. Therefore, it is assumed that probiotics protecting animals may also protect the humans who consume them. Many tested strains of probiotic bacteria can be used for individualized therapy. The recently isolated *Lactobacillus plantarum* R2 Biocenol™ has proven to be preferential in aquaculture, and potential benefits in humans are expected. A simple oral dosage form should be developed to test this hypothesis by a suitable preparation method, i.e., lyophilization, allowing the bacteria to survive longer. Lyophilizates were formed from silicates (Neusilin^®^ NS2N; US2), cellulose derivates (Avicel^®^ PH-101), and saccharides (inulin; saccharose; modified starch^®^ 1500). They were evaluated for their physicochemical properties (pH leachate, moisture content, water absorption, wetting time, DSC tests, densities, and flow properties); their bacterial viability was determined in conditions including relevant studies over 6 months at 4 °C and scanned under an electron microscope. Lyophilizate composed of Neusilin^®^ NS2N and saccharose appeared to be the most advantageous in terms of viability without any significant decrease. Its physicochemical properties are also suitable for capsule encapsulation, subsequent clinical evaluation, and individualized therapy.

## 1. Introduction

Currently, methods for increasing human immunity to resist exogenous agents such as pathogenic microorganisms are being sought [1,2]. At the same time, there has recently been growing concern about anthropozoonoses. Humans do not have the immune memory to cope with pathogens that are generally not found in their natural environment but commonly occur in animal organisms [3,4]. One of the prevention possibilities is the administration of probiotics, which increase resistance to these exogenous pathogens [5,6]. For example, different strains of lactobacilli are commonly used in veterinary and human clinical practice [5,7,8]. Some lactobacilli even show high adaptability to their surrounding conditions. They probably spread to animal species by the animals consuming the plant food that forms their native environment [9]. There are excellent studies on administering various strains of *Lactobacillus plantarum* of different origins to domestic pigs, significantly increasing their resistance to pathogens [10,11]. In addition, in aquaculture, *Lactobacillus plantarum* immunizes the intestines of commonly consumed young trout [12]. Due to the sustained effort to reduce meat production from livestock [13], compensation could be expected in the form of increased meat production from the fish industry [14]. As pathogens that are commonly found and cause comparable infections [15,16] in both fish and humans are well documented [17], it is appropriate to focus on probiotics that protect fish populations for eventual human administration, with the intention of also having the same potential positive impact on humans [18]. Expanding the spectrum of probiotics to include lactobacilli derived from other animal species would enhance human immunity to exogenous pathogens transmitted to humans from animals [19]. As animal and human probiotics are used not only to prevent and treat infectious diseases but also, with positive effects, to treat wound inflammation or to improve tissue healing [20,21], the administration of probiotics, including lactobacilli, could be used for treatment and prevention through individualized therapy in human or veterinary medicine [22,23,24,25]. Lactobacilli, when used for prevention and treatment, are stored in established microbial banks [26,27,28] and could be used when needed [29,30]. In addition, this could be applied to the microorganisms present in trout aquaculture [31]. In particular, *Lactobacillus plantarum* R2 Biocenol™ has shown antimicrobial susceptibility, and this was evaluated based on the guidelines issued by the European Food Safety Authority (EFSA). Furthermore, this probiotic strain has also shown inhibitory activity against common pathogens in aquaculture (*Aeromonas salmonicida* subsp. *salmonicida* and *Yersinia ruckeri*) [32]. Although these pathogens are expected to occur mainly in aquaculture, cases of infection with *Aeromonas salmonicida* and *Yersinia ruckeri* have also been reported in humans [15,16]. In addition, according to in vitro experiments, its resistance to different pH levels and temperatures, good growth characteristics, and survival were confirmed [33]. These microbes isolated from trout intestines could therefore be a source for the individualized preparation of specific medicinal forms with bacterial contents [34] for human treatment. Thus, there is an assumption that lactobacilli from a plant origin, which help animals to increase their resistance to plant pathogens [35], could also help humans to prevent or treat pathogenic diseases that are animal-transmitted to humans. The high risk of increased incidence of atypical infections in humans is thought to be due to the intensification of fish farming and the high levels of fish production for human consumption. Therefore, a similar effect in humans is expected based on the inhibitory activity of this strain against pathogens in aquaculture [12] and the anti-inflammatory effect [36] of this strain on fish. Hence, to protect human health against these pathogens, it is necessary to develop a new dosage form of *Lactobacillus plantarum* R2 Biocenol™ for humans. However, further clinical studies on human volunteers are needed to confirm this hypothesis. This work aimed to design a pharmaceutical dosage form containing *Lactobacillus plantarum* R2 Biocenol™, isolated from the intestine of rainbow trout (*Oncorhynchus mykiss*), suitable for clinical use in the human population. *Lactobacillus plantarum* strains appear more stable in the distal parts of the digestive tract [37,38]; therefore, the prerequisite is developing a technology and dosage form to administer viable lactobacilli to more distant parts of the human intestine without their denaturation in the stomach’s acidic environment or due to the influence of small-intestinal proteases [39]. The acid and bile tolerance of probiotic strains in conditions similar to a fish and human body were already tested in another experiment related to this bacterial strain [12]. The principle could be a bacterial lyophilizate in a powder form containing *Lactobacillus plantarum* R2 Biocenol™ as a model bacteria encapsulated in enteric capsules suitable for an individualized dosage preparation form for small patient cohorts for immediate use in a clinical study [40,41]. A composition must be found to create a suitable powder lyophilizate, making it possible to obtain a homogeneous material for encapsulation while allowing the bacteria to survive in the required time interval. Insoluble silicates [42] and cellulose derivates [43], used as constitutional excipients, and water-soluble polyols [44] or saccharides [45], which primarily act as cryoprotectants, could be promising for the formation of the required lyophilizates [46]. Hence, this work deals with the formulation and evaluation of a total of 16 lyophilizate compositions containing *Lactobacillus plantarum* R2 in mixtures with magnesium aluminosilicates (Neusilin^®^ US2 (US2); Neusilin^®^ NS2N (NS2N)), microcrystalline cellulose (Avicel^®^ PH-101 (AV)), and saccharides (inulin (IN), saccharose (SA), and modified starch 1500^®^ (ST)). The result was a lyophilizate formed as a solid dispersion containing lactobacilli. These solid dispersions were essentially analogous to the so-called co-process excipients (CPE) [47] containing silicates [48], cellulose derivates [49], or polyols [50]. As this is the same application, the standard CPE evaluation methodology of the physicochemical properties of powders was used to evaluate the lyophilizates. In addition to the viability of the bacteria and the residual moisture content, these lyophilizates have not yet been evaluated in this way in terms of these parameters (pH leaching, water absorption, wetting time, densities, and flow properties), so this was a new approach for lyophilizates in drug form. Differential scanning calorimetry tests and scanning samples under an electron microscope were performed.

## 2. Materials and Methods

### 2.1. Materials

MRS broth, MRS agar (both Biolife Italiana S.r.l., Milano, Italy), and sodium chloride (Slavus, Bratislava, Slovakia) were used in the cultivation and examination processes of the bacteria. MRS broth and MRS agar were prepared according to the manufacturer’s technical sheet [51]. Neusilin^®^ US2, Neusilin^®^ NS2N (magnesium aluminosilicates, Fuji Chemical Industries Cp., Ltd., Toyama, Japan), and Avicel^®^ PH-101 (microcrystalline cellulose, FMC Biopolymer, Wallingstown, Ireland) were used as constitutional excipients for the formation of the lyophilizate. In addition, inulin (Glentham Life Sciences, Ltd., Corsham, UK), saccharose (Sigma-Aldrich, St. Louis, MO, USA), and starch 1500^®^ (Colorcon^®^, Limited, Dartford, UK) were used as cryoprotectants in the lyophilization process. Various experiments used purified water (prepared in-house) as a solvent. Methylene blue (Fagron a.s., Olomouc, Czech Republic) was used in the determined evaluation.

### 2.2. Cultivation and Preparation of Probiotic Bacteria Suspension

For the preparation of the bacterial suspension intended for lyophilization, the probiotic strain *Lactobacillus plantarum* R2 Biocenol™ (CCM 8674) (recently taxonomically called *Lactiplantibacillus plantarum* [52]) was selected. Lactobacillus strain was first isolated from the intestines of rainbow trout (*Oncorhynchus mykiss*) raised on a commercial fish farm, i.e., Rybárstvo Požehy s.r.o., in the Slovak Republic [12]. Then, the probiotic strain was isolated and tested at the University of Veterinary Medicine and Pharmacy (Košice, Slovak Republic) and subsequently sent to the Czech Collection of Microorganisms (CCM) of Masaryk University (Brno, Czech Republic) for patent procedure purposes under the Budapest Treaty [53]. First, the bacterial culture was prepared by incubating 10 mL of inoculated MRS broth for 18 h at 37 °C. Subsequently, a 100 mL bacterial culture was prepared by adding 90 mL of pure MRS broth and incubating at the same conditions (18 h; 37 °C). In the last step, 100 mL of the prepared culture was transferred into 900 mL of the same pure MRS broth and incubated (18 h; 37 °C). After the last step, the bacterial culture was divided into 50 mL centrifuge tubes (Ecomed, Žilina, Slovakia) and centrifuged in a Boeco C-28A centrifuge (Boeckel & Co., Hamburg, Germany) at 4500 rpm (15 min; 22 °C). After centrifugation, the supernatant was removed, and 1 mL of 0.9% solution of sodium chloride was added to each pellet. The final step was to shake the resulting bacterial suspension and add the volumes of the centrifuge tubes into the resulting product.

### 2.3. Preparation and Lyophilization of Samples with Bacterial Culture

Pre-prepared bacterial culture was mixed with various types of cryoprotectants (IN; SA; ST) and different types of constitutional excipients (US2; NS2N; AV), always in the same ratio of 8:1:1 (bacterial culture: constitutional excipient: cryoprotectant) under aseptic conditions in a Labox BHL 65 laminar box (LABOX, spol. s.r.o., Jirny u Prahy, Czech Republic). For comparison, samples with one component (cryoprotectant or constitutional excipient) were prepared under similar conditions with a preserved ratio of components. In addition, a batch without any additive part (pure culture of lactobacilli bacteria) was prepared as a blank for comparison in the evaluation step. Each batch comprised four samples weighing 20.0 g per sample in separated lyophilization containers. Then, all the samples were deeply frozen at −80 °C in an ultra-low-temperature Arctiko ULTF 320 freezer (Arctiko, Esbjerg, Germany) for at least 2 days. The next step of the experiment was a lyophilization process (freeze-drying) using an L4-55 PRO lyophilizer (GREGOR Instruments s.r.o., Říčany, Czech Republic) precooled at −40 °C. The containers with samples were opened and covered with perforated aluminum foil to prevent sample loss. Firstly, the lyophilization was carried out until the probe and sample temperature evened out (36 ± 2 h in vacuum conditions) and then for another 4 h to ensure secondary drying. After lyophilization, the samples were closed and wrapped with parafilm (Bemis, Sheboygan Falls, WI, USA) to prevent access to air and moisture and were stored in a CALEX Symphony 340 refrigerator (Samsung Electronics Co., Suwon, Republic of Korea) in the dark at a temperature of 4 ± 0.5 °C within a relative humidity of 41 ± 1%. The characteristics of the composition of all 16 prepared batches are shown in Table 1.

### 2.4. Weight Loss of Samples after the Freeze-Drying Process

All batches were weighed twice, before and after the lyophilization process, on a KERN 870–13 analytical scale (Gottl. KERN & Sohn GmbH, Balingen, Germany). The average weight loss of the batches (%) ± SD after the lyophilization process was calculated based on the difference in weight before and after the process. The difference in weight indicated the weight loss (WL) of the water content of the lyophilized samples (%).

### 2.5. Standard Methods for Evaluation of Physicochemical Properties of Powder

Before evaluation, all samples were disaggregated to a powder form by a moderate pressure in a ceramic smooth-walled mortar and pestle under the same standard conditions (time, temperature) to achieve a uniform particle size and to preserve the bacterial structure in the lyophilizate. They were then stored at a temperature of 2–8 °C. This preparation method was chosen considering the preparation of individualized therapy for encapsulation into capsules.

The pH values from a 2.0% leachate of the lyophilized samples were determined three times using a surface pH microelectrode connected to an HI2211 pH/O.RP pH meter (HANNA^®^ Instruments, Praha, Czech Republic). Mean values ± SD were calculated.

Moisture content (MC) expressed as a percentage in the samples was assayed in an HX204 halogen moisture analyzer (Mettler Toledo, Zürich, Switzerland) under a drying temperature of 105 °C. A total of 0.5 g of the samples was used for each of the three measurements. The results were expressed as mean values (%) ± SD.

The wetting time (WT) and water absorption ratio (WA) of the lyophilized samples were determined according to a modified procedure [49]. First, a sponge (5.0 × 5.0 cm) was placed into a Petri dish and impregnated with 15.0 g of a 0.2% solution of methylene blue in purified water for easier identification of the complete wetting of the lyophilized samples. Next, 0.1 g of the lyophilized samples was weighed into a 3D-printed circle (1.0 × 0.5 × 1.0 cm; internal diameter of 0.5 cm) and carefully placed on the surface of the impregnated sponge in the Petri dish. The time required for the complete wetting of the sample (WT) by the color solution was noted (see Figure 1). The weight of the sample in the dry state was noted as *m*_0_ using a KERN 870–13 analytical scale (Gottl. KERN & Sohn GmbH, Balingen, Germany). Finally, the wetted samples were carefully removed and reweighed to obtain *m*_1_. *WA* was calculated using the following Equation (1):*WA* = 100 × [(*m*_1_ − *m*_0_)/*m*_0_](1)

The measurements were repeated three times, and the mean values of WT (s) and WA (%) ± SD were determined.

The lyophilized samples’ true density (g/mL) was determined under gas conditions using a Pycnomatic-ATC helium pycnometer (Porotec GmbH, Verder Scientific, Hofheim am Taunus, Austria) according to Ph. Eur. 2.9.23. [54]. All measurements of pycnometric density (PD) were performed in triplicate. The results were expressed as mean values (g/mL) ± SD.

The bulk and tapped densities (BD; TD) were determined using an SVM 102 tapping device (Erweka GmbH, Langen, Germany) apparatus according to Ph. Eur. 9 with the cylinder size adjusted to 10.0 mL due to the smaller sample amounts. From these volumes, the bulk and tapped densities (g/mL) were calculated, which was followed by the determination of the Hausner ratio (HR) and Carr’s compressibility index (CI) according to Ph. Eur. 2.9.36. [54]. All measurements were expressed as mean values from three measurements ± SD.

### 2.6. Differential Scanning Calorimetry (DSC) Measurement

DSC was performed using a DSC 7 testing machine instrument (PerkinElmer Instruments^®^, Waltham, MA, USA). The heating rate and flow rate were calibrated at 10 °C/min using indium and zinc standards. The heat flow rate was set at 10 °C/min, and an inert nitrogen atmosphere (0.35 MPa) was employed. Approximately 5 mg of every sample was weighed in a vented aluminum pan with a crimp-on lid. All samples were analyzed over a temperature range of 50–250 °C except the LP_IN and LP_SA samples.

### 2.7. Determination of Bacterial Culture Viability

The viability of the probiotic strain of Lactobacillus plantarum R2 Biocenol™ in every prepared lyophilized sample (see Table 1) was evaluated in triplicate immediately after lyophilization and during the following five months (with a total of six evaluations for every batch). A reduced temperature of 3–5 °C is optimally used for storing lactobacilli strains [55], so a storage temperature of 4 °C was chosen based on previous studies with the same type of bacteria [56], and the shortest limit of 6 months was chosen for the viability testing [57]. Therefore, storage conditions were as follows: inert glass vials sealed with parafilm in a dark area at a temperature of 4 ± 0.5 °C with a relative humidity within 41.0 ± 1%. As this study aimed to develop a dosage form suitable for encapsulation in enteric capsules and the viability of this bacteria strain in acidic conditions has already been tested [12], only the pH value of the lyophilizate leachate (such as those in intestinal conditions) was considered in this experiment.

The evaluation was performed by the plate method on MRS agar. First, weighted lyophilizates were diluted in a 0.9% sodium chloride solution by decimal dilution. Subsequently, MRS plates were inoculated with 100 μL of the prepared suspension. Cultivation of the probiotic strain lasted 48 h at 37 °C in anaerobic conditions in an AnaeroGen anaerobic box (Thermo Scientific, Cambridge, UK). The number of bacteria was expressed as a logarithm of colony-forming units per milliliter (log CFU).

The results of the bacterial viability analysis were statistically evaluated in GraphPad Prism using two-way ANOVA /one-way ANOVA and an additional Tukey’s post hoc test/Dunnett’s test. The results were evaluated in significance levels: *p* ˂ 0.05, 0.01, and 0.001 indicated the viability plots, respectively.

### 2.8. Macroscopic and Microscopic Images of Lyophilized Product

Macroscopic images of the lyophilizate cake were obtained from the above immediately after the lyophilization process with a D5500 digital camera (Nikon, Tokyo, Japan) equipped with an AF-S DX Micro NIKKOR 40 mm f/2.8G lens (Nikon, Tokyo, Japan).

To observe the structure and morphology of the lyophilizate cake and when trying to discover intact lactobacilli, all samples were also subjected to examination under a scanning electron microscope (SEM; MIRA3, Tescan Orsay Holding, Brno, Czech Republic) equipped with a secondary electron detector. A part of the lyophilized cake was mounted on an SEM stub (carbon double-faced adhesive tape—Agar Scientific, Essex, UK) and coated with gold (20 nm layer, argon atmosphere—Q150R ES Rotary-Pumped Sputter Coater/Carbon Coater, Quorum Technologies, Laughton, UK). SEM images were obtained through a secondary electron detector (an accelerating voltage of 3 kV with various image view fields, and the images of the view fields at 20 µm and 1000 µm are included in this article).

### 2.9. Statistical Analysis

Multivariate analysis of obtained results was carried out using principal component analysis (PCA) on standardized data. Mutual relationships between input variables (formulation parameters) and output variables (selected characteristics of lyophilizates) were assessed using two PCA models. As formulation parameters, the type of constitutional excipient and cryoprotectant were considered. The following variables were selected as the monitored outputs: physicochemical properties of the samples, bacteria viability at 0 and 6 months, and viability maintenance calculated as a percentage of the viability at 6 months relative to that at 0 months. Firstly, a PCA model, including all the samples, was created, and then the PCA model after the LP sample exclusion was built. A second PCA model was created to include the water absorption in the analysis (because of the NA value of this variable for the sample LP) and for comparison with the first PCA model. A deeper understanding of the relationships and the statistical significance of the individual effects was investigated using ANOVA (the significance level was considered to be 0.05). Pearson’s correlation coefficient was used to evaluate the association among the selected lyophilizate properties. The R software [58], versions 4.2.1 and 3.6.0, was used for data processing.

## 3. Results and Discussion

This experimental study prepared 16 batches of samples for the lyophilization process with different compositions of excipients and cryoprotectants. Subsequently, the samples were subjected to a lyophilization process and evaluated to investigate whether this chosen procedure for the preparation of probiotic lyophilization was suitable in terms of the continued viability of the contained bacteria and to determine which combination of excipients/cryoprotectants was the most suitable for this process and had the potential for further encapsulation in enteric capsules for individualized therapy. The best lyophilizate composition was selected by determining the viability of the bacteria. Then, the critical physicochemical parameters influencing the viability results were determined following the statistical analysis, and stability tests of the selected lyophilizate were carried out after 6 months of storage for the given parameters.

### 3.1. Weight Loss of Samples after the Freeze-Drying Process

According to the obtained results (see Table 2), after the lyophilization process, the weight of all the samples decreased approximately by more than 50.0%, from 59.98 ± 1.88 (LP_NS2N_ST) to 94.93 ± 0.18% (LP). Adding constitutional excipients to the mixture decreased the weight loss of samples since these substances remained inert [59,60,61] during the lyophilization process and thus formed the basis for the lyophilization cake, in which the presented bacteria could be better stabilized.

### 3.2. Results of Standard Evaluation of Physicochemical Properties of Powder

An acid–base balance is crucial for proper bacterial growth; however, lactobacilli are generally the most acidophilic of the lactic acid bacteria [62], even though samples with pH values ranging from 3.68 ± 0.00 (LP_IN) to 3.71 ± 0.01 (LP) had no optimal viability results (See below in Section 3.4. Determination of bacterial viability). *Lactobacillus plantarum* shows potent Growth in pH conditions around 4.2–4.5 [63,64]. Leaching from the samples containing US2 and NS2N with pH values between 4.88 ± 0.01 and 5.13 ± 0.01 seemed to influence the viability of the bacteria after the rehydration of the lyophilizate (see Table 2). The MC generally describes the residual moisture in samples [65] and is one of the main factors indicating solid drugs’ stability and mechanical properties in powder form [66]. The values for all the tested lyophilized powder samples ranged between 0.41 ± 0.09 (LP) and 4.28 ± 0.17% (LP_US2). Following the results of similar studies, it was determined that a value of MC < 5.0% in all the lyophilized samples would predict their stability [67,68]. The two parameters, WT and WA, are closely related to the hydrophilicity of the substances in the sample composition [69]. High WT values from 0.43 ± 0.12 to 11.43 ± 1.25 s (see Table 2) indicated that the lyophilized samples would react rapidly upon contact with GIT fluids. The high solubility and structure of SA [54,61] probably led to its possible loss during the process (see Figure 1), which may be a consequence of the low WA values in the samples containing this cryoprotectant (see Table 2). The results showed statistically significant differences between the samples depending on their composition. From a biopharmaceutical point of view, differences in WA and WT would not influence the release of bacteria from the dosage form. Concerning the requirements of Pharmacopoeia, an immediate-release dosage form should disintegrate within 15 min [54], which implies a very rapid release of the drug. Lyophilizates are generally used to ensure rapid release [62]; thus, the short WT times of the lyophilized samples (see Table 2) would provide this requirement. In this particular application, the flow properties and density of the powdered lyophilizates could influence the encapsulation of the material into the enteric capsules. The PD values in the range of 1.61 ± 0.01–2.24 ± 0.01 determined the true density of the samples (see Table 2). Based on the TD and BD values, the HR and CI values were calculated to characterize the lyophilized material. According to the limits of Pharmacopoeia [54], the powder mixtures had different flow properties (see Table 2). In general, the lyophilized samples could thus be distinguished into samples with average flow properties (LP_IN: HR 1.29 ± 0.01; CI: 22.38 ± 0.83) and samples with excellent flow properties (LP_US2: HR 1.05 ± 0.02; CI: 4.99 ± 1.65). In addition, the samples containing the cryoprotectant SA achieved the best flow properties in all the samples compared to the other cryoprotectants (see Table 2), so it was assumed that its content improved the flow properties of the lyophilized mixtures and, thus, improved its suitability for encapsulation. Bacterial lyophilizates are usually prepared with the addition of saccharides [70,71], which serve as cryoprotectants and, due to their solubility, form a homogeneous and compact lyophilization cake [72,73]. However, the resulting lyophilization cake would not be as compact if combined with insoluble inert metasilicates or cellulose derivatives based on their properties [74,75,76], which have not yet been used for this type of application. Thus, adding CPE excipients did not create a compact lyophilization cake but produced lyophilization cakes that were more suitable for pulverization and subsequent encapsulation in enteric capsules. In this case, they also improved the stability and viability of the bacteria because they formed a lyophilizate base to which the bacteria could adhere; however, due to their inertness [75], it was assumed that they did not adversely affect the bacterial viability. In addition, the individual relationships between the given parameters are summarized more precisely in the statistical analysis below.

### 3.3. DSC Measurement

The obtained data from the DSC measurements showed no relevant glass transitions in any of the samples. Only minor differences were detectable regarding the effect of the filler type on the state of the individual cryoprotectants. The cryoprotectant IN (T_m_ = 106.2 °C) was mainly amorphous in the LP_AV_IN sample. Partial crystallinity was observed in the LP_US2_IN sample and was slightly more pronounced in the LP_NS2N_IN sample. As for SA (T_m_ = 146.1 °C), only the LP_US2_SA sample contained partially crystalline SA; otherwise, it was in amorphous form (see Figure 2). The DSC results indicated no significant crystalline changes in the prepared mixtures containing the bacteria that could interfere with the bacterial wall and viability.

### 3.4. Determination of Bacterial Culture Viability

The lyophilization process is the most common way to keep bacteria in an inactivated form to avoid losing colony-forming units (CFU) during a long storage period. This method also enables keeping adequate living probiotics for later use [77,78]. However, this process can damage the structure of probiotic bacteria cells (see Figure 3). Therefore, protective substances should be used with minimum risk to the prepared system and to provide protection during the freeze-drying process [45,79]. In addition, the optimal pre-freezing temperature and freeze-drying process differ for every bacteria [80]. Therefore, it is crucial to adapt these parameters of lyophilization to ensure the highest survival rate of the probiotic strain. Furthermore, studies have proven the positive efficiency of different substances as protective agents (cryoprotectants) [81,82,83]. So far, no literary record in the current register has directly compared the effect of different types of cryoprotectants (ST; SA; IN) with a combination of these constitutive excipients (AV; NS2N; US2) on the cryoprotection of probiotic strains.

In addition, the stability of such or similar lyophilized products during the freeze-drying process and subsequently over a 6-month storage period has not yet been studied in similar studies (see Table 1 and Figure 3). Based on the viability results after the freeze-drying process (at time 0), ST showed the best cryoprotective effect (see Figure 3A–C) when all the tested samples showed a higher number of CFU than the LP (pure sample without any protective substance) at time 0. However, during storage, there was a significant decrease in the number of lactobacilli bacteria. According to the results, pure IN without any additives in the composition as a cryoprotectant was not able to preserve a high number of living bacteria, and the numbers of CFU were significantly lower than in the case of the pure LP sample (see Figure 3G–I), even though the prebiotic and cryoprotective potential was assumed [81,83,84]. However, by adding a constitutional excipient, especially NSN2N, a synergic effect with IN was confirmed in the LP_NS2N_IN sample (see Figure 3H), but a significant decrease in the number of bacteria occurred during the storage period (*p* ˂ 0.001; *p* ˂ 0.01). There were no more considerable differences between the mixtures of the constitutional excipients with the bacterial culture.

Regarding combinations of AV, NS2N, and US2 with the cryoprotectants and bacterial culture, the most stable was the LP_NS2N_SA sample, which presented a high number of CFU of *Lactobacillus plantarum* R2 Biocenol™ after lyophilization at time 0 (9.81 log CFU/mL, respectively), with no significant change in the number of bacteria even after a 6-month storage period at time 6 (9.72 log CFU/mL, respectively) (see Figure 3E). A good outcome was also demonstrated with sample LP_US2_SA, but a statistically significant decrease (*p* < 0.05) in the CFU of the bacteria was noted at the end of the storage period at time 6 compared to time 0 (see Figure 3F). This viability result in the combined mixture was comparatively better than the results of a study with pure SA added as a cryoprotectant [72]. Even though the LP_AV_ST and LP_AV_IN samples had promising results after freeze-drying at time 0 (9.19 and 9.71 log CFU/mL, respectively), a significant difference in the number of probiotic bacteria was noted after 3 months of storage, with a continuing decreasing trend in time being observed (see Figure 3A,G). Ultimately, the obtained outcomes from the viability measurements showed that the mixtures of the constitutional excipients NS2N or US2 with the cryoprotectant SA could be considered to be a suitable combination for protection during the lyophilization process and in the preservation of a stable number of probiotic strains in mixtures for a longer time (minimum of 6 months) with optimal storage conditions (dark area, 4 ± 0.5 °C; 41 ± 1% relative humidity). The bacterial strain used in this experiment showed a high survival rate in the stomach-like conditions (pH = 2.0; 2.5; 3.0) [63,64] occurring in the host organism *Oncorhynchus mykiss* with an optimal body temperature at 12 °C [85] and at 37 °C (human body temperature). The viability of more than 90.0% of the bacterial cells was noted at pH values of 2.5 and 3.0. A similar survival rate was recorded after 4h of exposure in the presence of 10.0% trout bile [12]. These findings are fundamental for the further transition of probiotics through the stomach environment and the subsequent adhesion and colonization of intestinal mucus. Although acid and bile tolerances are essential, different probiotic strains show the highest viability at enteric or neutral pH [86,87]. Therefore, to protect and avoid potential losses of viable probiotics at low pH, it could be assumed that the encapsulation of bacterial lyophilizates into enteric capsules could be a suitable dosage form for subsequent administration into the intestine.

### 3.5. Macroscopic and Microscopic Images

The lyophilization process generally creates products with a higher porosity because of the sublimation of water that has been deeply frozen and forms cavities in the resulting product [88]. However, the macroscopic images of the lyophilization cakes (see Figure 4A1–A4) with constitutive excipients produced more compact homogeneous cakes than the pure sample or the samples with the addition of a cryoprotectant alone. There were also differences between the types of cryoprotectants. For example, the most compact lyophilized cake was formed by IN or the sample with ST; conversely, the products with SA were more porous and more likely to be prone to a collapsing lyophilized cake [73]. The SEM images obtained in the 1000 µm view field (see Figure 4B1–B4) showed the various structures and morphologies of the constituent excipients (spherical particles of NS2N and US2 and the flake-like structure of AV) [75,89]. In addition, the effect of the added cryoprotectant on the structure is also displayed (see Figure 4). The microscopic images showed well the fact that the moderate disaggregation of the lyophilizate into a powder form resulted in only those particles corresponding to the lyophilization cake (see Figure 4). According to the obtained SEM images, the lactobacilli structures with a body size of 3–8 µm [90,91] were visible in all the samples with the addition of a constitutional excipient (see Figure 4C1,C2) in the 20 µm view field.

### 3.6. Statistical Analysis

Visualizing the PCA results via scores and loadings plots revealed two fundamental data correlation trends. Both the PCA models—the PCA model including all the samples (see Figure 5A,B) and the PCA model excluding the LP sample (see Figure 5C,D)—showed a similar data correlation structure. Therefore, for the sake of simplicity, the discussion related to the first model will be conducted below.

#### 3.6.1. Group of Variables: pH, PD, MC, HR, and Via 0

In the direction approximately from the upper right quadrant to the lower left quadrant, the effect of the constitutional excipient could be observed in the scores plot; the distribution of samples gradually from AV to NS2N/US2 (see Figure 5A). When the combination of these findings and the loadings plot (see Figure 5B) was considered, it was deduced that, compared to the samples with AV, the samples containing NS2N/US2 were characterized by higher Via 0, pH, PD, and MC levels and, at the same time, lower HR levels.

Statistically significant differences between the SA and NS2N/US2 sample groups were confirmed through ANOVA for all these quantities (*p* < 0.05 in all cases). In addition, in some cases, a significant mutual difference between the NS2N and US2 samples was also found (US2 > NS2N for pH and MC; *p* < 0.05).

The differentiation of the samples based on the cryoprotectant type and the interactions was also determined to be statistically significant for almost all the mentioned features of the lyophilizates (*p* < 0.05 in all cases except for the pycnometric density, where the effect was insignificant). However, the influence of the cryoprotectant type on the value of the mentioned variables was considerably lower than the excipient effect. This fact was visible due to the distribution of the objects in the ordination space of the first two principal components and in the one-dimensional data visualizations, which were not included in this article. Moreover, it was impossible to define any general unequivocal trend for these quantities regarding the effect of the cryoprotectant type.

#### 3.6.2. Group of Variables: WL, WA, BD, TD, Via 6, and Via 6_0

Based on the effect of the cryoprotectant type, a second essential correlation trend was manifested in an approximately perpendicular direction in the PCA model. The SA samples were located in the upper left quadrant, and the samples containing IN occupied the space around the PC1 axis. The ST samples lay below them in the PCA ordination space, more to the right, and were followed by the LP sample (see Figure 5A). In the given order (SA–IN–ST), a change in the following quantities was observed: an increase in the weight loss and a decrease in Via 6, Via 6_0, BD, and TD (see Figure 5B). Based on the second PCA model (see Figure 5C,D), it can be said that the WA also increased in the order of the samples SA–IN–ST.

ANOVA, in conjunction with basic data visualizations, confirmed the following differences based on cryoprotectant type: ST > IN > SA for WA; SA > IN > ST for Via 6, BD, and TD; SA > ST/IN for Via 6_0 and IN/SA > ST for WL (*p* < 0.05). However, as was apparent, only the WL quantity trend did not precisely correspond to the PCA visualization (see Figure 5), which was probably caused due to the higher interaction effects. Moreover, the second PCA model (see Figure 5C,D) showed a meager contribution of this quantity to the explained variability (a short vector). The effect of the excipient and the interaction between the substance type and cryoprotectant type were also confirmed to be statistically significant for all the variables in this group (*p* < 0.05 in all cases). However, in contrast to these effects, the influence of the cryoprotectant type was crucial. The LP sample could be distinguished from the other samples mainly in the direction of the WL arrow. The significantly higher WL and lower MC values for LP compared to the other samples were confirmed using ANOVA (*p* < 0.05 for both cases).

#### 3.6.3. Correlation Analysis

A correlation analysis was performed to examine the interrelationships of pairs of selected variables (see Table 3). A statistically significant positive correlation was revealed between the following variables: BD and TD; Via 0, Via 6, and pH; Via 6 and Via 6_0; MC and Via 0; and pH and MC. Conversely, a negative correlation was confirmed for the following quantities: WA and BD/TD; HR and Via (in all cases: time 0 and 6 months as well as viability maintenance); PD and TD; and pH and HR (see Table 3). From a formulation perspective, the most relevant parameter was Via, which positively correlated with two factors, namely pH and HR. As demonstrated in previous studies, a higher pH value is unfavorable for the survival of lactobacilli [64]. The HR value gave us the difference between BD and TD due to the spaced air responsible for the inter-particular porosity. This air contains oxygen, which partially damages anaerobic bacteria, including lactobacilli [37,64]. The observed positive correlation of both values was, therefore, a confirmation of previous conclusions.

Based on the viability results, which correlated with the pH and HR parameters, the stability tests of the most advantageous composition after a 6-month storage period with the constitutional excipient NS2N and cryoprotectant SA were performed. No considerable changes in the physicochemical parameters influencing the viability (assessment of leachate pH and determination of HR) of the samples were detected during the 6-month storage period ((HR at time 0 was 1.12 ± 0.02; at time 6 it was 1.12 ± 0.01; pH at time 0 was 5.02 ± 0.01; pH at time 6 was 5.04 ± 0.02). Therefore, according to the obtained stability results, it was assumed that chosen composition was even suitable for a long-term storage period at a suitable temperature of 4 °C concerning the relevant study [55].

## 4. Conclusions

This experiment aimed to develop the technology and composition for preparing a pharmaceutical formulation containing the probiotic bacteria strain *Lactobacillus plantarum* R2 Biocenol™, isolated from the intestines of rainbow trout (*Oncorhynchus mykiss*), for the prevention of potential anthropozoonoses and the treatment of certain inflammatory diseases. Since the crucial aspect of the formulation is the viability of the bacteria, lyophilization with selected conditions was chosen as an appropriate gentle preparation method. At the same time, it was required to determine a suitable composition of the lyophilizates using constitutional excipients from a range of cellulose derivatives (Avicel^®^ PH-101) or silicates (Neusilin^®^ NS2N; Neusilin^®^ US2) in combination with adequate cryoprotectants from selected polysaccharides (inulin; saccharose; modified starch 1500^®^). The composition of the lyophilizate containing the excipient Neusilin^®^ NS2N and the cryoprotectant saccharose was found to be the most suitable in terms of viability, with no significant decrease being observed after a storage period of 6 months, which was chosen as the time limit based on the literature. Furthermore, even this composition had advantageous physicochemical properties without any considerable change after the 6-month storage period at an optimal temperature of 4 °C for the intended encapsulation in capsules. The result was a biologically stable solid bacterial dispersion in the form of a lyophilizate that could be used for encapsulation into hard enteric capsules that would protect the bacteria from stomach acid and proteases, allowing the individualized preparation of an enteric dosage form under laboratory conditions that would be ready for immediate use. Furthermore, the bacterial viability assays showed that the experiment’s aim was fully attained for the Neusilin^®^ NS2N and saccharose sample, which yielded the complete preservation of cell viability for more than 6 months.

## Figures and Tables

**Figure 1 pharmaceutics-15-00910-f001:**
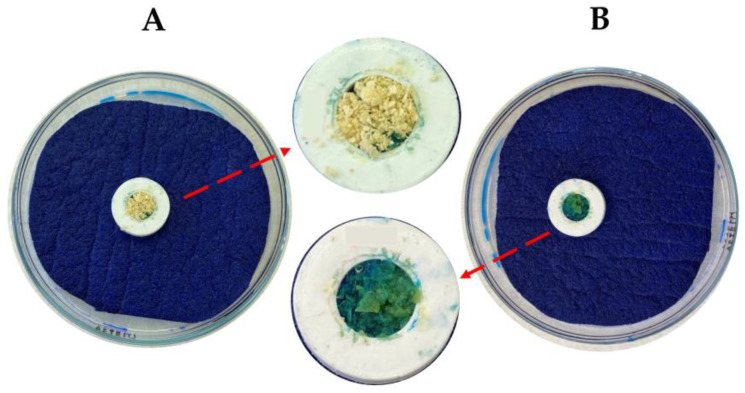
Determination of WT in two steps: (**A**) before wetting and (**B**) complete sample wetting.

**Figure 2 pharmaceutics-15-00910-f002:**
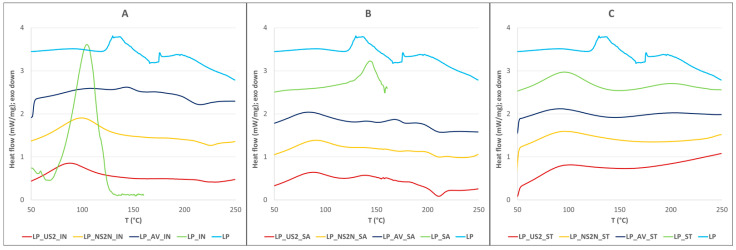
DSC plots of lyophilized samples with different cryoprotectants compared with a pure lyophilized sample: (**A**) IN; (**B**) SA; and (**C**) ST.

**Figure 3 pharmaceutics-15-00910-f003:**
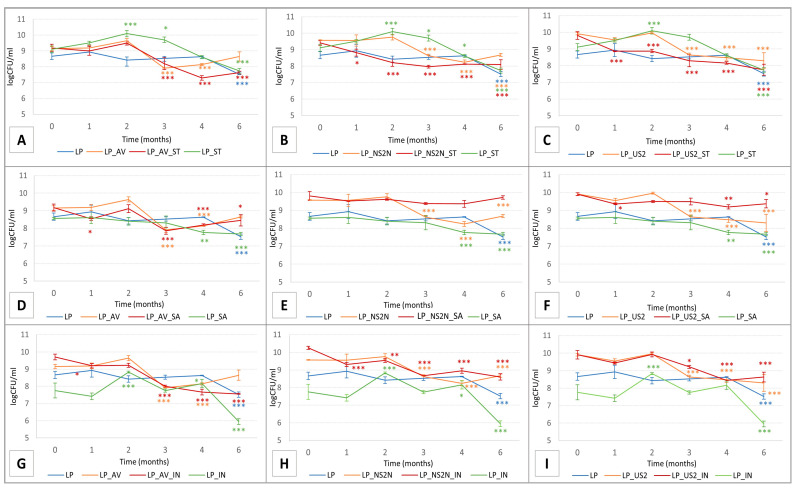
Comparison of viability plots of lyophilized samples containing Lactobacillus plantarum R2 Biocenol™ in different composition combinations of constitutional excipients and cryoprotectants: starch ^®^ 1500 with: (**A**) Avicel^®^ PH-101, (**B**) Neusilin^®^ NS2N, and (**C**) Neusilin^®^ US2; saccharose with: (**D**) Avicel^®^ PH-101, (**E**) Neusilin^®^ NS2N, and (**F**) Neusilin^®^ US2; inulin with: (**G**) Avicel^®^ PH-101, (**H**) Neusilin^®^ NS2N, and (**I**) Neusilin^®^ US2. Statistical significance from two-way ANOVA test with additional Dunnett’s multiple comparison tests is marked with a color resolution for each curve as follows: *** *p* < 0.001; ** *p* < 0.01; * *p* < 0.05, respectively.

**Figure 4 pharmaceutics-15-00910-f004:**
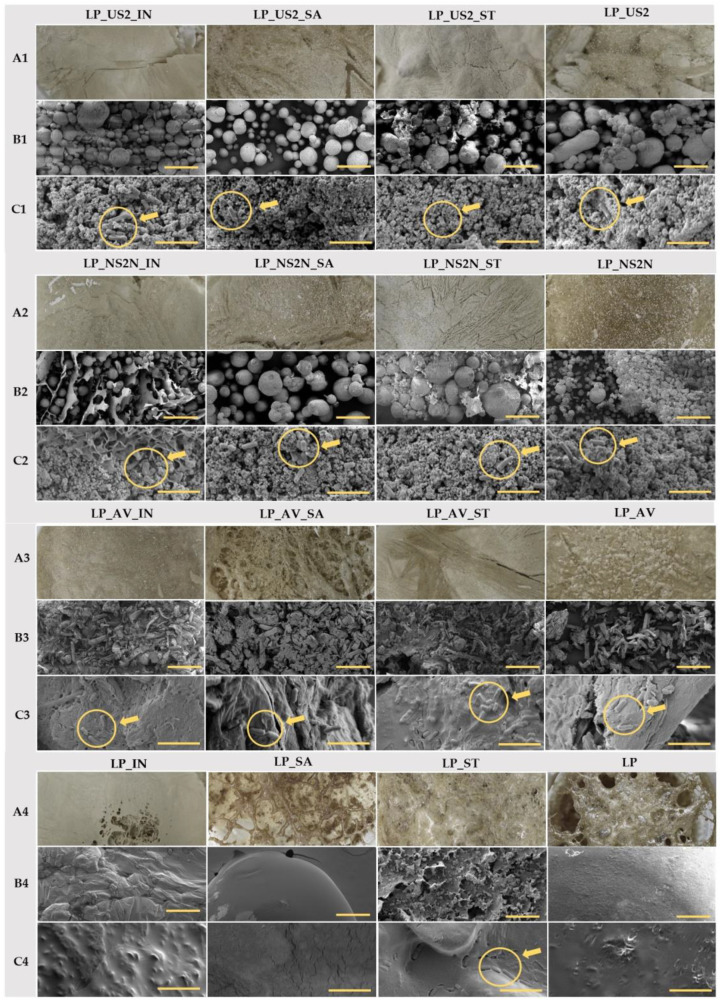
Lyophilized samples containing bacteria in three different types of images: (**A**) macroscopic; (**B**) SEM images in the view field of 1000 µm with a bar corresponding to 200 µm; and (**C**) SEM images in the view field of 20 µm with a bar equivalent to 5 µm. The circle with a pointing arrow directly indicates the lactobacilli body.

**Figure 5 pharmaceutics-15-00910-f005:**
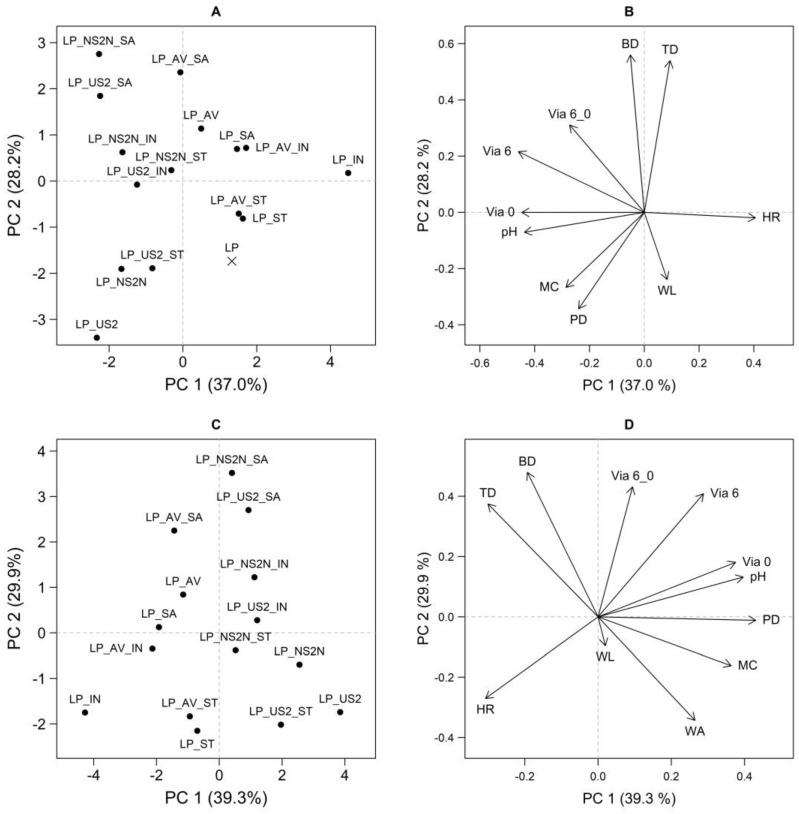
PCA scores plot and loadings plot for model including all samples (**A**,**B**) and model after exclusion of LP sample (**C**,**D**). Variables: weight loss (%); pycnometric density (PD); bulk density (BD); tapped density (TD); Hausner ratio (HR); pH leaching (pH); moisture content (MC); water absorption ratio (WA); and viability at time 0 month (Via 0), 6 months (Via 6), and viability maintenance (Via 6_0).

**Table 1 pharmaceutics-15-00910-t001:** Summary of prepared bacterial batches (abbreviation; composition of batches).

Abbreviation of Prepared Batch ^1^	Components of Batches ^2^		
	Constitutional Excipient	Cryoprotectant	Bacterial Culture
LP_US2_IN	Neusilin^®^ US2	Inulin	*Lactobacillus plantarum* R2 Biocenol™
LP_US2_SA		Saccharose	
LP_US2_ST		Starch 1500^®^	
LP_NS2N_IN	Neusilin^®^ NS2N	Inulin	
LP_NS2N_SA		Saccharose	
LP_NS2N-ST		Starch 1500^®^	
LP_AV_IN	Avicel^®^ PH-101	Inulin	
LP_AV_SA		Saccharose	
LP_AV_ST		Starch 1500^®^	
LP_IN	–	Inulin	
LP_SA		Saccharose	
LP_ST		Starch 1500^®^	
LP_US2	Neusilin^®^ US2	–	
LP_NS2N	Neusilin^®^ NS2N		
LP_AV	Avicel^®^ PH-101		
LP	–	–	

^1^ Abbreviation of each sample was established by the presence of *Lactobacillus plantarum* R2 Biocenol™ bacterial culture (LP); constitutional excipient (Neusilin^®^ US2 (US2); Neusilin^®^ NS2N (NS2N); Avicel^®^ PH-101 (AV)); and cryoprotectant employed (inulin (IN); saccharose (SA); starch 1500^®^ (ST)). ^2^ Each batch consisted of quadruple samples (weighing 20.0 g per sample). Components (bacterial culture: constitutional excipient: cryoprotectant) were always prepared in the preserved ratio of 8:1:1.

**Table 2 pharmaceutics-15-00910-t002:** Weight loss (WL) after the freeze-drying process; physicochemical properties of the lyophilized sample (pycnometric density (PD); bulk density (BD); tapped density (TD); Hausner ratio (HR); Carr’s compressibility index (CI); pH leaching; moisture content (MC); wetting time (WT); water absorption ratio (WA)) ± calculated SD.

Sample	WL (%)	PD (g/mL)	TD (g/mL)	BD (g/mL)	HR	CI	pH	MC (%)	WT (s)	WA (%)
LP_US2_IN	75.25 ± 1.77	1.84 ± 0.00	0.32 ± 0.00	0.38 ± 0.00	1.19 ± 0.01	16.06 ± 0.75	5.13 ± 0.01	2.75 ± 0.54	0.77 ± 0.25	167.79 ± 4.87
LP_US2_SA	79.09 ± 3.97	1.79 ± 0.00	0.40 ± 0.00	0.44 ± 0.00	1.10 ± 0.01	8.85 ± 0.90	5.02 ± 0.01	2.40 ± 0.77	1.67 ± 0.58	97.55 ± 8.63
LP_US2_ST	70.78 ± 1.54	1.80 ± 0.07	0.26 ± 0.01	0.30 ± 0.00	1.16 ± 0.04	13.69 ± 3.27	5.03 ± 0.00	3.61 ± 0.65	5.67 ± 0.58	299.75 ± 6.75
LP_NS2N_IN	79.49 ± 2.99	1.84 ± 0.01	0.38 ± 0.02	0.43 ± 0.01	1.13 ± 0.04	11.49 ± 3.41	5.01 ± 0.01	2.85 ± 0.13	7.53 ± 0.50	127.01 ± 9.25
LP_NS2N_SA	71.85 ± 2.24	1.75 ± 0.00	0.41 ± 0.01	0.46 ± 0.00	1.12 ± 0.02	10.83 ± 1.44	5.02 ± 0.01	1.85 ± 0.79	2.60 ± 0.53	55.49 ± 2.22
LP_NS2N_ST	59.98 ± 1.88	1.81 ± 0.00	0.30 ± 0.00	0.34 ± 0.00	1.16 ± 0.00	13.89 ± 0.00	4.90 ± 0.01	1.04 ± 0.30	2.70 ± 0.27	287.92 ± 5.17
LP_AV_IN	77.64 ± 2.37	1.55 ± 0.00	0.35 ± 0.01	0.44 ± 0.00	1.25 ± 0.02	20.17 ± 1.26	3.68 ± 0.01	2.24 ± 0.32	11.43 ± 1.25	64.62 ± 7.93
LP_AV_SA	74.49 ± 2.28	1.65 ± 0.00	0.41 ± 0.01	0.44 ± 0.00	1.08 ± 0.02	7.53 ± 1.88	3.68 ± 0.00	1.55 ± 0.24	3.93 ± 0.12	39.35 ± 6.86
LP_AV_ST	76.74 ± 0.82	1.59 ± 0.00	0.27 ± 0.00	0.34 ± 0.00	1.23 ± 0.02	18.79 ± 1.28	3.72 ± 0.01	2.42 ± 0.26	5.33 ± 0.15	219.13 ± 6.93
LP_IN	80.50 ± 0.57	1.61 ± 0.01	0.34 ± 0.00	0.44 ± 0.01	1.29 ± 0.01	22.38 ± 0.83	3.68 ± 0.01	1.30 ± 0.57	2.77 ± 0.25	24.70 ± 7.19
LP_SA	76.90 ± 1.30	1.64 ± 0.01	0.32 ± 0.00	0.36 ± 0.01	1.13 ± 0.03	11.43 ± 2.47	3.68 ± 0.00	0.57 ± 0.30	0.77 ± 0.25	88.54 ± 4.41
LP_ST	81.76 ± 1.27	1.64 ± 0.01	0.27 ± 0.01	0.34 ± 0.00	1.25 ± 0.06	20.07 ± 3.73	3.69 ± 0.01	2.29 ± 0.36	7.83 ± 0.76	307.32 ± 6.28
LP_US2	86.00 ± 0.69	2.06 ± 0.15	0.23 ± 0.01	0.24 ± 0.01	1.05 ± 0.02	4.99 ± 1.65	4.88 ± 0.01	4.28 ± 0.17	0.70 ± 0.30	226.58 ± 7.02
LP_NS2N	82.94 ± 2.68	2.06 ± 0.00	0.25 ± 0.00	0.29 ± 0.00	1.15 ± 0.01	12.90 ± 0.69	5.00 ± 0.00	2.84 ± 0.58	0.60 ± 0.17	191.83 ± 1.10
LP_AV	82.43 ± 1.51	1.68 ± 0.02	0.34 ± 0.00	0.41 ± 0.01	1.22 ± 0.02	17.65 ± 1.62	3.74 ± 0.01	1.92 ± 0.11	0.43 ± 0.12	94.83 ± 6.01
LP	94.93 ± 0.18	2.24 ± 0.01	0.28 ± 0.00	0.33 ± 0.00	1.17 ± 0.00	14.29 ± 0.00	3.71 ± 0.01	0.41 ± 0.09	N/A *	N/A

* N/A—not applicable.

**Table 3 pharmaceutics-15-00910-t003:** Correlation matrix of selected variables: Pearson’s correlation coefficients, significant correlation indicated in bold.

Variable	WL	pH	MC	WA ^1^	PD	BD	TD	HR	Via 0	Via 6	Via 6_0
WL		−0.33	0.02	−0.21	0.47	−0.25	−0.19	0.05	−0.22	−0.17	−0.03
pH			**0.56 ***	0.31	0.44	−0.01	−0.20	**−0.51 ***	**0.74 ****	**0.61 ***	0.21
MC				0.44	0.15	−0.32	−0.39	−0.24	**0.68 ****	0.28	−0.21
WA ^1^					0.37	**−0.78 ****	**−0.79 *****	0.02	0.27	−0.05	−0.29
PD						−0.41	**−0.56 ***	−0.46	0.22	0.22	0.13
BD							**0.94 *****	−0.15	0.10	0.38	0.45
TD								0.20	−0.07	0.13	0.24
HR									**−0.50 ***	**−0.66 ****	**−0.51 ***
Via 0										**0.75 ****	0.21
Via 6											**0.80 *****

^1^ Correlation coefficients for all combinations were calculated based on all samples except for combinations involving water absorption, where values for samples 1–15 were used (NA value for water absorption of sample 16). Statistical significance was as follows: *** *p* < 0.001; ** *p* < 0.01; * *p* < 0.05, respectively.

## Data Availability

The datasets corresponding to the current study are available from the corresponding author upon request.

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
