# Peer review of "Preparation and Evaluation of a Dosage Form for Individualized Administration of Lyophilized Probiotics"

_pharmaceutics, 2023, doi:10.3390/pharmaceutics15030910_

Round 1
Reviewer 1 Report
Lyophilization of several formulations of Bacillus plantarum were compared aiming at long-term survival of the bacteria for further use as probiotics. The excipients were silicates (Neusilin® NS2N; US2), cellulose derivatives (Avicel® PH-101), and saccharides (inulin; saccharose; modified Starch® 1500). The cell viability assays showed that the aim was fully attained for Neusilin® NS2N and saccharose that yielded complete preservation of cell viability for more than 6 months.
Reviewer 2 Report
1. The results and discussion section does not have a discussion part. It must be included separately with a lot of relevant references.
2. The abstract should contain some discussion and conclusion parts.
3. The conclusion must exclude excess results and it should have discussion.
4. The manuscript is not organised as per mdpi formatting.
Reviewer 3 Report
Tha manuscript has described the work on probiotic preparation by lyophilized process. The work is intersting but the result should be carefully presented and discussed. The comments are;
1. It will be better if the author could separate the result into 2 parts; Lyophilization process and 6 months storage process. The analysis data of product after lyophilization were presented but it would be better it the data of storaged product(after 6 months storage) were presented too.
2. More discussion on the effect of the Constitutional excipient and Cryoprotectant on bacterial cell during lyophilization process and storage should be added.
3. The criteria for the optimal conditon was not clearly presented and discussed.
4. The new finding of this work should be highlighted.
Reviewer 4 Report
The work by Fulupova et al. describes an experimental study of the lyophilization of a Lactobacillus plantarum strain from trout's intestine. The strain was selected for its potential to enhance human immunity and for its inhibitory activity against common pathogens in aquaculture. The study evaluates the physicochemical properties of lyophilized samples, including weight loss, moisture content, wetting time, water absorption ratio, and true density. The viability of the probiotic strain in each sample was also evaluated. The study uses principal component analysis to assess the mutual relationships between formulation parameters and selected characteristics of lyophilizates. The study concludes that adding constitutional excipients to the mixture decreases weight loss and that lyophilized samples would react rapidly upon contact with gastrointestinal fluids. Prior to publication, however, the authors need to address the following concerns:
1. The introduction of the manuscript presents a common problem (anthropozoonoses) and proposes a potential solution (probiotics). The idea of using lactobacilli from other animal species as probiotics for humans is not entirely new, but the application of a specific strain isolated from trout and its potential use in aquaculture is somewhat novel. Overall, the introduction provides a good background and rationale for the study, but it does not introduce any groundbreaking concepts. Please include them.
2. The DSC plots are hard to see and failed to show the important thermal transitions.
3. One important experiment is missing where cell viability is explored in different simulated gastrointestinal buffers. This is key to determine whether under conditions similar to those observed physiologically, the microorganisms are able to survive. Please include such experiments and their results.
Round 2
Reviewer 2 Report
Authors have addressed my comments.
Reviewer 3 Report
The manuscript is much improved and it can be accepted. However, please the quality of figure should be improved with higher resolution.